# Social Perception of the Ecosystem Services of *Prunus serotina* subsp. *capuli* in the Andes of Ecuador

**Juan Carlos Carrasco Baquero** [1], **Luis Felipe Lema Palaquibay** [2], **Carlos Renato Chávez Velásquez** [1,*], **Verónica Caballero-Serrano** [1], **Rachel Itle** [3] **and Dario J. Chavez** [3,4]

1 Faculty of Natural Resources, Escuela Superior Politécnica de Chimborazo (ESPOCH), Riobamba 060155, Ecuador; juancarlos.carrasco@espoch.edu.ec (J.C.C.B.); veronica.caballero@espoch.edu.ec (V.C.-S.)
2 CAPULI—Investigation Project, Escuela Superior Politécnica de Chimborazo, Ecuador & Investigation Institute, Riobamba 060155, Ecuador; luisf.lema@espoch.edu.ec
3 Department of Horticulture, University of Georgia, Griffin, GA 30229, USA; ritle@uga.edu (R.I.)
4 Institute of Plant Breeding, Genetics and Genomics, University of Georgia, Griffin, GA 30229, USA
* Correspondence: renato.chavez@espoch.edu.ec

**Abstract:** Ecosystem services (ES) refer to the benefits that people obtain from the ecosystem. In this sense, *Prunus serotina* is associated with the provision of ES; however, these services have been scarcely studied. The objective of this research was to determine the knowledge and perceptions of individuals in rural areas regarding the importance of ES, as well as the factors that influence them. Surveys were applied in three study areas (Chimborazo, Tungurahua, and Cotopaxi) of the central Andes of Ecuador that detailed the sociodemographic and perception characteristics of ES based on the predefined ES of the Millennium Ecosystem Assessment (MEA). In the assessment, the interview data were analyzed to obtain the relationships between the variables using Spearman's correlation in the R-studio software. The results showed that individuals' level of education, age, and gender play an important role in variations in peoples' knowledge of ES. A total of 21 ES were identified; the most representative services, according to their ecosystem category, were support (shelter for birds and insects and soil formation), provisioning (food, insecticide, wood, and firewood), regulation (improvement in the quality of air and climate regulation), and culture (scenic beauty and the maintenance of traditions). This analysis of the social perception of ES works as a strategy for the maintenance of *Prunus serotina* in the orchards and plots of families in the central Andes of Ecuador. The identification of ES through the social perception of their existence facilitates an understanding of the importance of ES in rural localities, which lays the foundation for strategies to be developed in the future for their conservation.

**Keywords:** social perception; ecosystem services; *Prunus serotina*; sociodemographic; capuli; Andes





## 1. Introduction

Ecosystem services (ES) allow for an analysis of the existing links between ecosystems and human well-being [1]. The degradation of ecosystems has drawn the attention of the international scientific community, giving rise to studies such as the Millennium Ecosystem Assessment (MEA), which revolves around provisioning, regulating, and supporting ES, as well as their cultural impact. The objective of this assessment is to evaluate the functions of ecosystems in human well-being; additionally, it aims to generate a scientific basis for the conservation and sustainable use of these systems [2]. Its importance lies in the conditions, processes, and functions that characterize natural ecosystems and their biodiversity [3,4]. The loss of these will bring with them economic and social costs due to the problems caused in the production of goods and services, thus affecting or decreasing the impact of so-called ES. On the other hand, Robertson and Swinton [5] highlighted that active management

of the provision of multiple ES could substantially reduce the environmental footprint of agriculture.

Popenoe and Pachano [6] highlighted that capuli is a species that has been distributed throughout the American continent, from Canada to southern Bolivia. In South America, this species is characterized by the large size of its fruit (2.0 to 3.5 cm) and its pleasant flavor. It is likely that varieties of capuli with large fruit were the result of domestication processes. Based on the selection of this species by the flavor, size, and quality of its fruit [7], it is of special interest since these species have a great antioxidant capacity [8] and a high content of minerals and proteins, such as hyperoxide, which generates antioxidant, vasodilator, and antihypertensive effects, characteristics that could potentially be useful in the prevention and treatment of hypertension [9].

Ecuador is a "mega diverse" country both biologically and culturally due to the significant variety of its ecosystems [10]. This is thanks to the particular geographical conditions of the country's location, relief, and climate [11]. The conjunction of these diversities has resulted in numerous useful flora based on the practice and management of agroecosystems, which are one of the main components that allow us to analyze rural territories with the aim of accelerating the conservation and development effort [12].

In Ecuador, capuli is distributed in dry areas along the inter-Andean alley, from 1800 to 3400 m.a.s.l. from the province Carchi, located in the northern limit, to the province Loja in the south of the country. However, there is a greater concentration in the provinces of the central Andes (Chimborazo, Tungurahua, and Cotopaxi) [13,14], where the ideal territory for the optimal development of this species is considered to be located [15].

Capuli is a tree whose existence is declining to a great extent. There are areas where the species is about to disappear [16] due to the poor use of land, for example, by turning land into public parking lots; in addition, many capuli trees have been cut down to make way for pastures. Above all, the species have been neglected by society due to phasing out, due to the rise of crops such as kidney tomato [17]. In terms of its population, productivity, yield, and product quality, Avendaño et al. [18], Tamayo et al. [16], Chisaguano [19], and Central Ecuatoriana de Servicios Agrícolas (CESA) [20] stated that in recent years, the population of trees of the *Prunus serotina* species in Ecuador has declined its presence by up to 57%.

Another factor that affects the reduction of the *Prunus serotina* species is the low viability of its seeds, since without any previous germination treatment, the germination percentage is less than 33.3% [21]. Germination occurs in the first or second year; there are times when germination is delayed up to 3 years, and on many occasions, these seeds lose their germination capacity because they do not find the appropriate environmental conditions [22].

Focusing on development species that have intrinsic value to the population is an effective way of conserving nature [23–25]. Primack [26] mentions that species of flora or fauna that have striking characteristics, such as the beauty of their flowers, and the specific use of some part of a plant or large mammals are frequently used to get people's attention and thus raise economic capital to save endangered species [27]. Sometimes it does not necessarily have to be an endangered species; many times they focus on the utility that a social group gives it, turning it into a symbolic species, which sometimes contributes to reinforcing the cultural associations of people with nature [28,29] that can encourage conservation interest in local natural environments [30].

Identifying the dependence of the local population on ES can become a baseline for the community management of the landscape [31]. In a study in the city of Sukuragawa, Japan, on the perception of the ES provided by wild cherry trees, the authors mention that for the conservation of such landscapes, it is important to disseminate knowledge about the perception of ES; in addition, they indicate that these sites should be prioritized for designation as protected areas and sites for community forest management [32].

According to Mouchet et al. [33] and Cord et al. [34], ES have proven to be useful for determining management decisions and environmental policies by means of analyzing their

impacts on the provision of ES. Data about the perception of ES by local residents are assets to develop strategies for the conservation and sustainable management of biodiversity in urban and peri-urban forests [35].

The main justification for this research lies in the fact that there are information gaps related to the ES provided by the *Prunus serotina* species in the central Andes of Ecuador. From this perspective, the social perception of the ecosystem services of *Prunus serotina* subsp. capuli in the Andes of Ecuador will be based on surveys and databases obtained in the field, for which the study aims to answer the following research questions: (1) How does the rural population of the central Andes of Ecuador identify and perceive the ES of the *Prunus serotina* species? (2) What are the importance and individuals' perceptions of changes in ES? (3) What sociodemographic factors determine the identification of ES? Finally, it is important to understand the possible implications of the results obtained from this research for the sustainable management of forests, as well as the design and implementation of conservation education in order to improve the attitudes of the population toward the management of natural resources.

## 2. Materials and Methods

This research was carried out in a location where *Prunus serotina* has a high distribution density, i.e., the Cotopaxi (CO), Tungurahua (TU), and Chimborazo (CH) provinces (Figure 1c) which are located in the mountainous systems of the central Andes of Ecuador (Figure 1b).

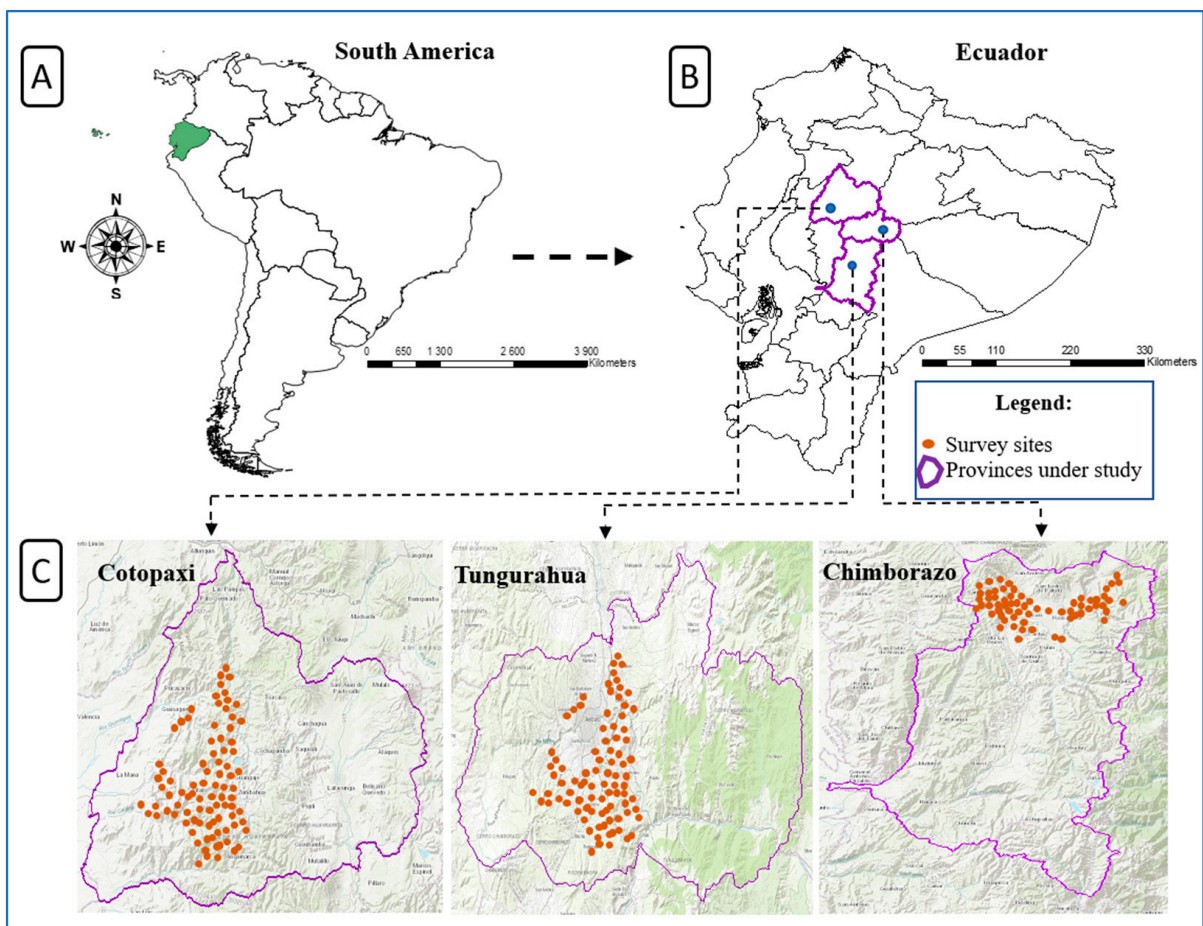

**Figure 1.** Map of the study area. (**A**) Location in relation to South America; (**B**) location in relation to Ecuador; (**C**) location of the Chimborazo, Tungurahua, and Cotopaxi provinces.

The data for this research were obtained from a face-to-face survey consisting of 7 questions (Appendix A) which were administered in December 2019 to a social group that comprised the population of the rural sector where *Prunus serotina* is distributed in the CO, TU, and CH provinces. The population was restricted to individuals older than 15 years of age because it is the population group that has the greatest amount of contact with the use and management of natural resources in the study area.

The population of this study was determined by considering the projection of the Ecuadorian population for the year 2019 from the 2010 census of the National Institute of Statistics and Censuses of Ecuador; for this, the population of the provinces of the central Andes of Ecuador where the species *Prunus serotina* is distributed was considered rural. There were 95,546 people in CO, 190,390 people in TU, and 157,459 people in CH [36].

The sample size of the population of the study for each province was calculated using the formula for finite populations (Table 1) [37].

**Table 1.** Values for the calculation of the sample.

| Formula | Variable | Cotopaxi | Tungurahua | Chimborazo |
|---|---|---|---|---|
| $n = \dfrac{N}{1 + \frac{e^2(N-1)}{z^2 \cdot pq}}$ | $N$ = universe | 95,546 | 190,390 | 157,459 |
| | $p$ = probability of occurrence | 0.5 | 0.5 | 0.5 |
| | $q$ = probability of non-occurrence | 0.5 | 0.5 | 0.5 |
| | $e$ = margin of error | 7% | 7% | 7% |
| | $z$ = confidence level | 1.96 | 1.96 | 1.96 |
| | $n$ = sample size | 196 | 196 | 196 |

The interviews were conducted orally; Spanish and colloquial language were used to explain the ES to the respondents. A total of 588 people completed the survey and were included in this study (response rate = 100%). Once the sociodemographic characteristics and perception of the ES were identified, the participants were asked to rate the importance of each service on a scale of 1 to 3 (with 1 being unimportant and 3 being very important) [38]. In addition, the trend of the supply of each of the ES was identified.

The relationship between the sociodemographic variables and the identified ES was determined by Spearman's rank correlation analysis [39], using R-Studio version 4.1.1 software [40].

## 3. Results

Of those who were interviewed, 60% were women, more than 33% of them were middle-aged (31–45 years old), and only 15% were over 60 years old. In addition, 91% of all those surveyed had attained at least one type of education level (Figure 2). Some of the interviewees were unaware of what ES are; however, after explaining what the term refers to, the residents were able to identify the services. Additionally, the participants mentioned the importance of the *Prunus serotina* species and the well-being it generates for human beings.

In the study area, through the application of the surveys, 21 ecosystem services were identified, which are distributed as supporting ES (14.3%), provisioning ES (28.6%), regulating ES (28.6%), and cultural ES (28.6%) (Figure 3). There are no differences between the number of ES identified in the provinces because they share similar uses and ancestral traditions among the Andean peoples.

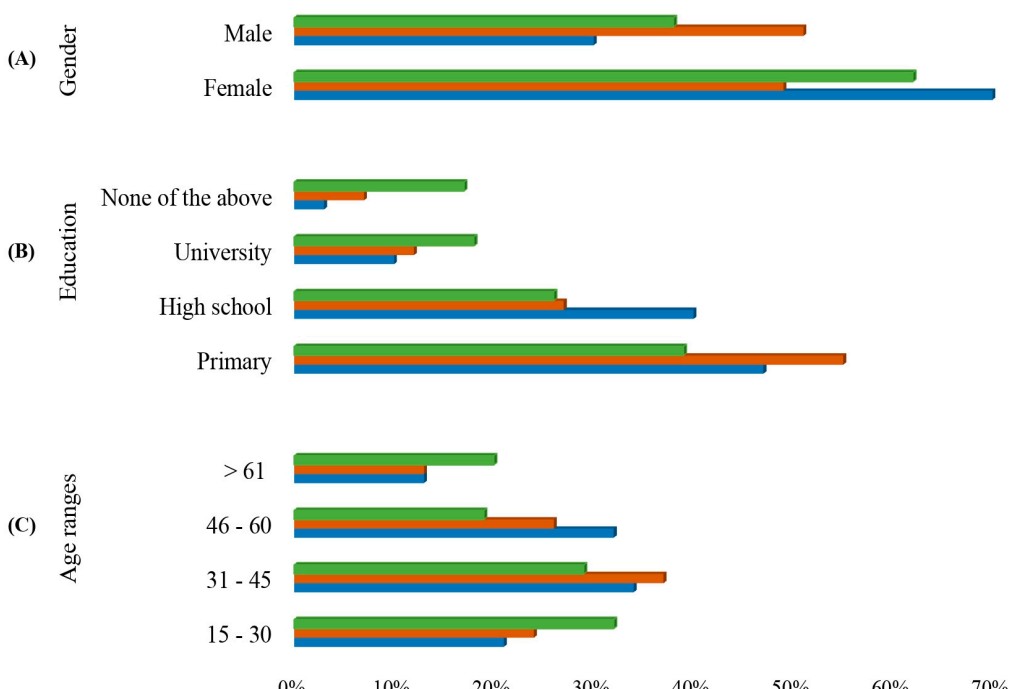

**Figure 2.** Sociodemographic characteristics. (**A**) Gender; (**B**) education; (**C**) age ranges. In the graph, the green (Chimborazo), orange (Tungurahua), and blue (Cotopaxi) colors correspond to the provinces under study.

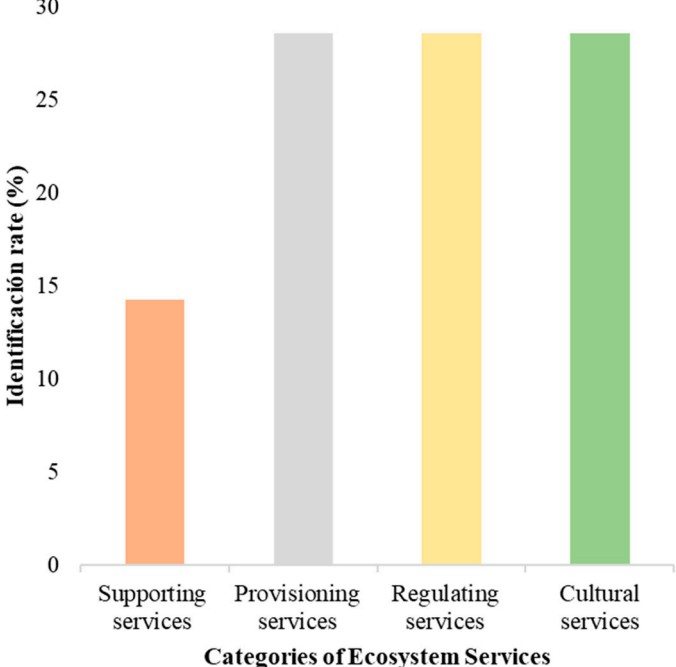

**Figure 3.** Identification of ecosystem services grouped by category.

### 3.1. Cultural Services

Six cultural ES were identified. The most frequently identified ES corresponded to the beauty of the landscape, with an occurrence of 16% in CO, 19% in TU, and 22% in CH, and the maintenance of traditions, corresponding to 51% in CO, 48% in TU, and 33% in CH, respectively (Figure 4); the latter showed significant differences between the localities studied. These results are in agreement with those reported by Calvet et al. [41] and Bernués et al. [42] who, in a study conducted in the central highlands of Mexico, determined

that the beauty of the landscape, relaxation, rest, and the maintenance of traditions were the ES most frequently identified. However, these differ from the study conducted by Martín et al. [43], who presented that the ES most frequently identified in their study were rituality and tranquility, which are frequently recognized and obtained higher values in the rural area, while other cultural services, such as scenic beauty and environmental education, were perceived more frequently in the urban area.

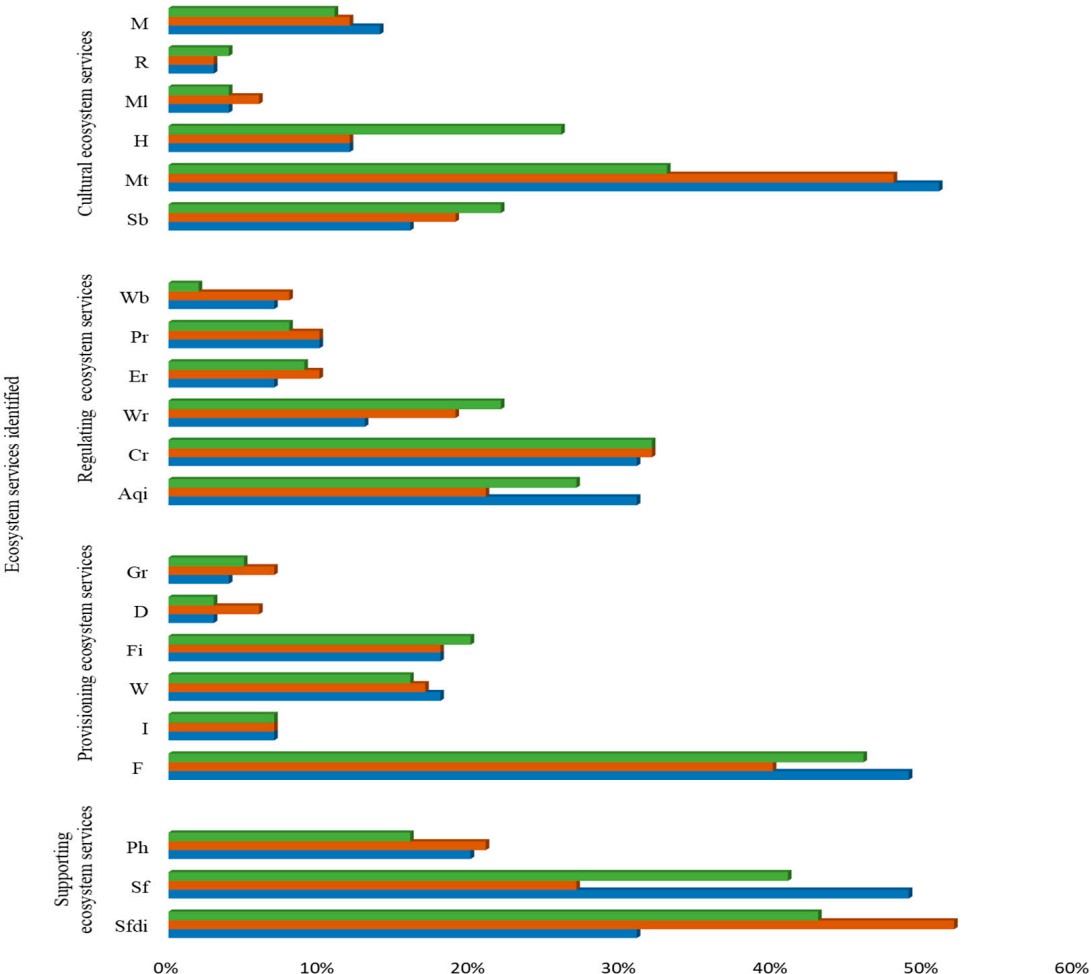

**Figure 4.** Identification rate of the ES provided by the species *Prunus serotina*, identified by the rural population of the localities (Chimborazo—green, Tungurahua—orange, and Cotopaxi—blue) located in the central Andes of Ecuador. Cultural ES: M: Medicinal; R: Recreation; Ml: Myths and legends; H: Handicrafts; Mt: Maintenance of traditions; Sb: Scenic beauty. Regulating ES: Wb: Windbreaker barrier; Pr: Pest regulation; Er: Erosion regulation; Wr: Water regulation; Cr: Climate regulation; Aqi: Air quality improvement. Provisioning ES: Gr: Genetic resources; D: Dyes; Fi: Firewood; W: Wood; I: Insecticide; F: Food. Supporting ES: Ph: Photosynthesis; Sf: Soil formation; Sfdi: Shelter for birds and insects.

### 3.2. Regulating Services

The regulating ES that prevailed in the provinces under study were an improvement in the air quality, with an occurrence of 31% in CO, 21% in TU, and 27% in CH, and climate regulation, corresponding to 31% in CO, 32% in TU, and 32% in CH (Figure 4). The level of education (Figure 2) had a positive influence on the identification of the regulating ES, as mentioned by Sodhi et al. [44]. According to Gouwakinnou et al. [38], the higher the education level, the higher the identification rate. Martin et al. [43] mentioned that rural populations more frequently identified provisioning ES, while urban populations mostly reported regulating services, since they contribute directly to the quality of life in an urban

context (air purification and regulation of the microclimate). De Groot et al. [45] determined that cities, as opposed to rural areas, are the most demanding of regulating services; this is because, in general, regulating services decrease with an increase in the intensive use of land. The difference in the results of the research carried out, compared to Martín and De Groot, is believed to be due to the accessibility to education and communication technologies, which the population of Ecuador has had in recent years as a national development objective [46].

### 3.3. Provisioning Services

In the CO, TU, and CH provinces, the provision of food, which corresponded to 49%, 40%, and 46%, respectively (Figure 4), ranked first in the prioritization of the environmental service of provisioning. The main utility the interviewees allocated to *Prunus serotina* fruit was the preparation of a typical drink known as "Jucho", which is a beverage that contains P. serotina fruit and peaches boiled together plus other seasonings. Palacios [47] mentioned that the native peasants of the inter-Andean region harvest capuli fruits for family consumption and commercialization at the local level; this is done only in the production season, corresponding to the months from December to March. According to Raya et al. [48], in their studies carried out in Mexico, they mention that capuli is part of the Mexican diet, and its fruits are popularly consumed when fresh, dried, or as ingredients in other dishes (such as in jellies, tamales, or liqueurs).

The interviewees also identified the ES of firewood and wood supply as representative; this is because firewood is used for cooking food, which is very common in the rural area of the Andes, and wood is used for carpentry because it is of good quality and is easy to handle [49]. According to Niembro et al. [50], the wood of *Prunus serotina* is used as a source of energy (firewood) or as a raw material to carry out various joinery works. According to Jiménez et al. [51], *Prunus serotina* wood has an important economic potential, characterized by its hardness, ease of handling, ability to acquire an attractive polish, and ability to be incorporated into the wood industry. As a forest species, this is a good species to obtain high-quality wood from [52].

For the identification of ES, the influence of the socioeconomic characteristics of the household also comes into play. This is generally known as the social "zone of influence" [53]. According to Gouwakinnou et al. [38], poor households tend to be more dependent on forest resources and may consider provisioning services to be the most important compared to rich households.

### 3.4. Supporting Services

The people interviewed identified three supporting ES; shelter for birds and insects was considered the most important because these are breeding, nesting, and feeding sites for various species of birds and insects, with an occurrence of 31% in CO, 52% in TU, and 43% in CH, respectively, for each province. In addition, the ES of soil formation and photosynthesis also occurred but less frequently (Figure 4). In this sense, the maintenance of high levels of biological diversity allows for a better performance of other ES, such as soil formation [54]. According to Englund et al. [55], support services are essential to maintain life conditions on earth and include services such as soil formation and photosynthesis. Therefore, the flow of ES determines the level of human well-being, which is linked to the composition and function of the ecosystem [56].

### 3.5. Perception of the Importance of ES and Their Trends

The population of the rural area of the three localities (CO, TU, and CH) of the central Andes of Ecuador identified the ES of soil formation (45%), food (43%), firewood (63%), and maintenance of the traditions (49%) as very important (Figure 5a). Camacho et al. [57] and Kimpouni et al. [35] mention that when local people are dependent on the natural environment for goods and income, their perception of ES tends to focus on provisioning services. The participants identified the ES of myths and legends (71%) and recreation (66%)

as not very important (Figure 5b), and they identified the ES of dyes (47%) and genetic resources (44%) as do not know (Figure 5c).

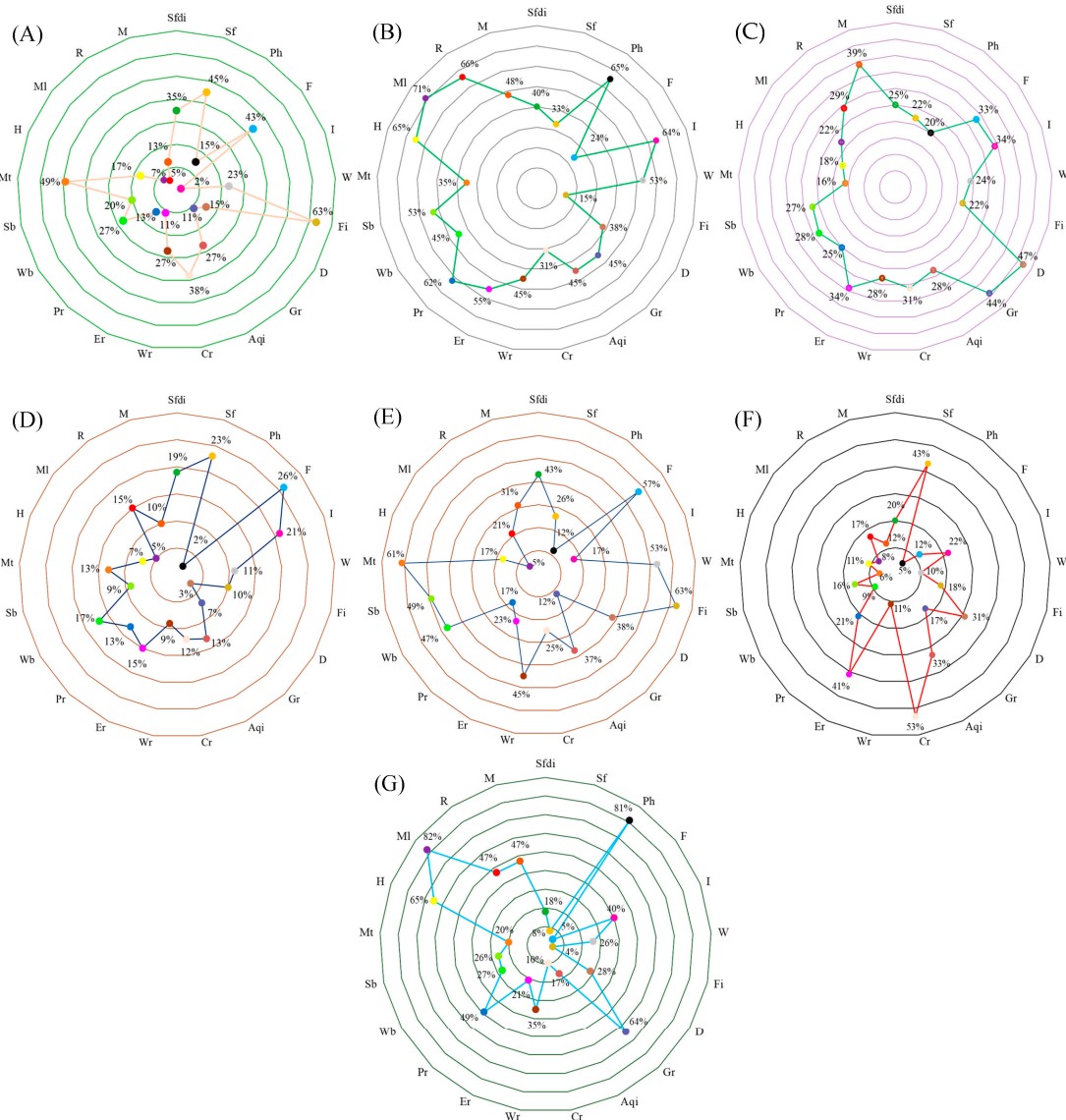

**Figure 5.** Perceived importance of ES: (**A**) very important, (**B**) not important, (**C**) do not know. ES trend: (**D**) improving, (**E**) declining, (**F**) no change, (**G**) do not know. Cultural ES: M: Medicinal; R: Recreation; Ml: Myths and legends; H: Handicrafts; Mt: Maintenance of traditions; Sb: Scenic beauty. Regulating ES: Wb: Windbreaker barrier; Pr: Pest regulation; Er: Erosion regulation; Wr: Water regulation; Cr: Climate regulation; Aqi: Air quality improvement. Provisioning ES: Gr: Genetic resources; D: Dyes; Fi: Firewood; W: Wood; I: Insecticide; F: Food. Supporting ES: Ph: Photosynthesis; Sf: Soil formation; Sfdi: Shelter for birds and insects.

In relation to the trends (improving, declining, no change, and do not know) of the ES, in the three study areas, the participants reported trends toward the reduction in ES, especially in services such as shelter for birds and insects (43%), food (57%), wood (53%), firewood (63%), the maintenance of traditions (61%), and water regulation (45%) (Figure 5e). The latter service may be due to periods of drought that have been periodic in Ecuador, the most prone areas being the inter-Andean region, which spans from the Carchi province to the Loja province [58].

The inadequate use of ecosystems to obtain one or more services generates an imbalance of these services [59]. An example of this is the intensification of agriculture in the

inter-Andean region, which seeks to satisfy the needs of the population but, in exchange, means reducing forest remnants in order to expand cultivated areas, thereby causing the loss of forests and biodiversity [60].

Identifying the priorities of society in terms of the identification, importance, and trend of ES contributes to the application of actions for its conservation, passive and active restoration, and responsible use for sustainable management by area. The perspective of rural inhabitants on ES should be considered in debates and decisions on policies related to the causes of environmental degradation due to land use change and the strategies to address them [59], which are key to maintaining the supply of ES [61] because changes in land use influence the supply of ES [45]. These results should contribute to decision making and planning on land use, aimed at improving human well-being and preserving ecosystem function, not only at the regional but also at the local level [62].

However, this study was conducted on a regional scale; on a local scale, there may certainly be differences in the strength of the relationships between the *Prunus serotina* species and ES. Therefore, all services may not be similarly maximized within the landscapes throughout the sample region. In addition, the present investigation focused on the current distribution of the highest concentration of the species *Prunus serotina*, but future changes in the environment may affect ES both directly and indirectly [63]. Therefore, global change may mean that society does not have all the ecosystem service management options currently available. The scenario of changes in the composition and richness of the *Prunus serotina* species due to global change must be considered in analyses of the future provision of ES [64].

### 3.6. Correlation Analysis

Some of the ES were positively correlated to each other (Figure 6), but some notable trade-offs could also be identified. Trade-offs between water regulation and climate regulation, bird sanctuary, and scenic beauty were first noted. Second, photosynthesis and firewood generation were negatively correlated.

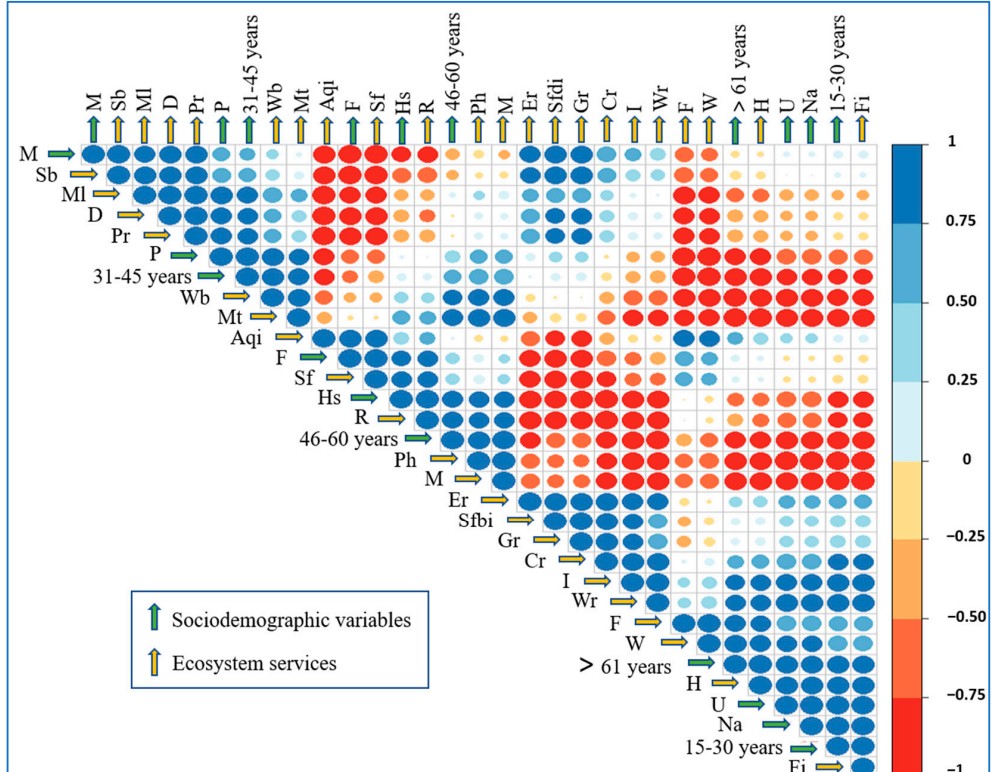

**Figure 6.** Correlation coefficient and its significance between sociodemographic variables (M: Male; F: Female; Na: None of the above; U: University; Hs: High school; P: Primary; 15–30 years; 31–45 years;

46–60 years; > 61 years) and cultural ES (M: Medicinal; R: Recreation; Ml: Myths and legends; H: Handicrafts; Mt: Maintenance of traditions; Sb: Scenic beauty), regulating ES (Wb: Windbreaker barrier; Pr: Pest regulation; Er: Erosion regulation; Wr: Water regulation; Cr: Climate regulation; Aqi: Air quality improvement), provisioning ES (Gr: Genetic resources; D: Dyes; Fi: Firewood; W: Wood; I: Insecticide; F: Food), and supporting ES (Ph: Photosynthesis; Sf: Soil formation; Sfdi: Shelter for birds and insects). Values close to 1 represent a higher correlation, values close to 0 indicate a lower correlation, and values close to −1 represent a negative correlation.

The relationship of the sociodemographic variables with ES showed a positive correlation between a higher education level and the identification rate of ES (soil formation, insecticide, air quality improvement, pest regulation, and recreation), concatenating with the above. According to Villamagua [65], people with a higher educational level more closely value the ecosystem service of provisioning. Knowledge influences behavioral attitudes and individual intentions [38]. For example, a farmer who has extensive knowledge of the consequences of insecticide use on insect populations (pollinators) [66] will develop behaviors to minimize their adverse effects [67].

People who do not have a higher level of education showed a positive correlation with the ES of food supply, firewood, water regulation, and handicraft production. Sodhi et al. [44] highlighted that people with a lower educational level, as well as poor people, value more the services (food, firewood, and wood) of forest ecosystems; this is related to what was described by Villamagua [65], where it was indicated that the highest percentage of interviewees mentioned that they were supplied with forest species for the construction of homes and for use as firewood. From a socioeconomic point of view, several studies have revealed that poor people value forest ES more, although they sometimes value them in different ways [44,68].

In relation to gender, there is a positive correlation between the male gender and regulation ES (erosion and pests) and cultural services (myths and legends); the female gender has a positive correlation with support services (photosynthesis) and provisioning (food and wood). Moutouama et al. [69] stated that gender affects the perception of ES; considering provisioning services, a higher percentage of women identified food and firewood, whereas in a study conducted by Villamagua [65], it was mentioned that men perceive provision services more frequently, while women perceive regulation services more often.

In relation to age, people between 15 and 30 years of age were positively correlated with the regulation of erosion and handicrafts; people aged 31 to 45 correlated well with myths and legends; people aged 46 to 60 were correlated with the maintenance of traditions and medicinal use; and those older than 60 years of age correlated positively with wood and firewood services. In relation to age, Briceño et al. [70] and Martin et al. [43] highlighted that younger people prioritize cultural services which deal with the regulation of the ecosystem and cultural ES more. According to Villamagua [65], those over 60 years of age value provisioning services more. These results coincide with those obtained in this study.

## 4. Discussion

Human beings constitute an integral part of ecosystems, but most of the time, they are not taken into account when conducting ES assessments [71]. In past years, understanding ES concepts related to user needs and the respective classification has not been adequately taken into account [72], improving such understanding can contribute to improving conservation efforts focused on ES [73]. The determination of the ES provided by the species *Prunus serotina* from a social perception (identification, importance, and tendency of the ES) can be constituted as a tool or a planning input for decision makers, who seek to identify priorities in planning processes for conservation in a territory [43,72]. Martín-López et al. [43] mention that in order to formulate successful policies for sustainable ecosystems, it is essential to understand the preferences, attitudes, and perceptions of the user toward

ES. In addition, studies involving understanding and data on the local social component of ES contribute to complementing national statistics on ES [74].

The implementation of ES-based research, both in policy and planning, demands the contribution of direct users, that is, local populations and other interested parties. This incorporation and input can contribute to improving the practical application and political relevance of SE concepts in management and decision making [71]. Individual and community preferences can coevolve with land management and land use change, but there is difficulty in demonstrating how a region's values, attitudes, and cultural norms contribute to ES preferences [75].

Societal preferences for ES vary across societies [38,76]. However, the most important services can be identified by taking into account the perspectives of stakeholders from different social groups [77]. These priority ES can be used for land planning and local development [78]. The provisioning and regulation ES were identified in a greater number and with a higher degree of importance in the three sites studied, which indicates that households depend especially on the provisioning services of the *Prunus serotina* species [79].

The importance of *Prunus serotina* as a provider of genetic resources was not identified, which suggests a lack of community participation in the conservation of the genetic material of fruit species in the central Andes of Ecuador. The soil formation ecosystem service was identified as relevant, which provides fundamental support for agricultural production; the opposite happened with the identification of the soil erosion regulation service, which was not very important for the studied localities, despite the predominance of the interviewed farmers.

The decrease in provisioning services, as reported by the residents of Chimborazo, Tungurahua, and Cotopaxi, was attributed to the misuse (construction of housing, highways, and whereabouts and areas of intensive cultivation) of the land, which causes the degradation or destruction of the habitat of this species, which does not allow for the sustainable management of this resource; therefore, it is important to implement awareness policies on good practices for the sustainable management of subsistence agroforestry resources in the local communities.

Cultural services, such as the maintenance of traditions and scenic beauty, were considered important in the three locations studied. According to Ahammad et al. [80], forests have a high cultural importance, which is reflected in the positive attitudes of the local population toward their protection. The perception of the population of the three locations under study regarding the value of ES should be considered when making management decisions for the *Prunus serotina* species.

Due to the diversity of ES identified by locals in the three study areas of the CO, CH, and TU provinces, there are opportunities to conserve specific sites in the central Andes of Ecuador in order to optimize the supply of ES for provisioning, regulating, supporting, and, above all, cultural services which focus on a cultural re-evaluation. This is because *Prunus serotina* plays an important role in the life of Ecuadorian people [81].

Most biodiversity conservation and restoration policies are based on the protection of species and habitats, which has led to the development of protection policies for certain geographical areas and the definition of protected areas (PA). These areas, with a special biodiversity value, are a very important base for conservation, yet most of the territory is unprotected, and much of the biodiversity is located outside the limits of these PA.

This implies that biodiversity conservation must encompass a broader territory than the protected areas and must be integrated into all aspects of human society. The protection of areas of high natural value, despite being necessary, is not enough to reduce the loss of biodiversity. In turn, conservation policies and the management of natural resources have been based on strategies aimed at planning in a given area without considering the possible global consequences of an activity on the environment and society [82].

If the aim is to minimize the loss of biodiversity worldwide in the coming decades, structural changes must be made in production and consumption patterns. In addition, an

integrated approach from different sectors is needed by combining different conservation measures [83]. Sustainable territorial planning implies integrating environmental, social, economic, and institutional aspects based on the recognition of the great interdependence between them [84]. This ecosystem-based management approach assumes a complete relationship between human well-being and the environment; sustainable development is only possible in both areas at the same time. This approach allows managers to gain a broad understanding of the many consequences of particular decisions [85]. Along with this idea, a renewed interest in biodiversity conservation strategies and policies must be developed in relation to the need to maintain a sustainable supply of ES.

The use of land to produce goods and services represents the most substantial human alteration of the Earth's natural systems [38]. The Millennium Ecosystem Assessment [2], in its evaluation report, determined that local ecological knowledge is relevant to address the problems of managing ES. Several authors have highlighted the fundamental importance of taking into account local knowledge and perceptions as a basic tool in decision-making policies for the protection of ecosystems and the sustainable management of resources and livelihoods [79,86–88].

The results obtained suggest that different types of knowledge may be required to identify the range of ES, such as experiential knowledge (comprises a set of non-scientific beliefs, knowledge, and practices corresponding to local ecosystems, and their management is based on the acquired local experience) and technical knowledge (constituted by a set of strict and universally accepted rules shaped by academic disciplines and scientific methods) [72,89–91]. Previous studies have shown that these two types of knowledge are complementary and that their combination can play an active role in the perception and maintenance of multiple ecosystem services provision [43,92,93]. According to Miller [94], identifying, evaluating, and employing multiple knowledge and learning mechanisms are the key tenets of sustainability science.

## 5. Conclusions

This study contributes to the understanding of the social perception of the ecosystem services of *Prunus serotina* in the central Andes of Ecuador. There is little research on the social perception of the ecosystem services of symbolic species; this is because most studies on the perception of ecosystem services focus on large-scale areas such as protected areas. In our study, factors such as the level of education, related to the level of both experiential and technical knowledge were essential in the identification of provisioning and support services. The three localities most frequently identified the importance and decline of provisioning services, mainly firewood because in the rural areas of the Andes of Ecuador, it is used as an ecological fuel for cooking food. Ecosystem service trade-offs occur as people modify landscapes due to their different perceptions, values, and interests. This study facilitates the understanding of the importance of ES in rural localities; this information should become an instrument for land use planning and the preservation of ecosystem functioning, in addition to establishing the basis for the design of future strategies focused on the conservation of the *Prunus serotina* species.

**Author Contributions:** Conceptualization, J.C.C.B., L.F.L.P. and V.C.-S.; formal analysis, J.C.C.B. and D.J.C.; investigation, J.C.C.B., D.J.C., R.I., C.R.C.V., L.F.L.P. and V.C.-S.; resources, J.C.C.B. and L.F.L.P.; writing—original draft preparation, J.C.C.B., D.J.C., R.I., C.R.C.V. and L.F.L.P.; writing—review and editing, J.C.C.B., D.J.C. and V.C.-S.; supervision, D.J.C.; funding acquisition, D.J.C. and J.C.C.B. All authors have read and agreed to the published version of the manuscript.

**Funding:** The research in Ecuador was funded by Escuela Superior Politécnica de Chimborazo (ESPOCH). The research in the US was funded by the University of Georgia through an inter-institutional collaborative research agreement.

**Informed Consent Statement:** Not applicable.

**Data Availability Statement:** Data are contained within the article.

**Acknowledgments:** The authors thank our collaborators at ESPOCH, as well as the technical assistance of the Department of Horticulture at the University of Georgia.

**Conflicts of Interest:** The authors declare no conflict of interest.

**Appendix A**

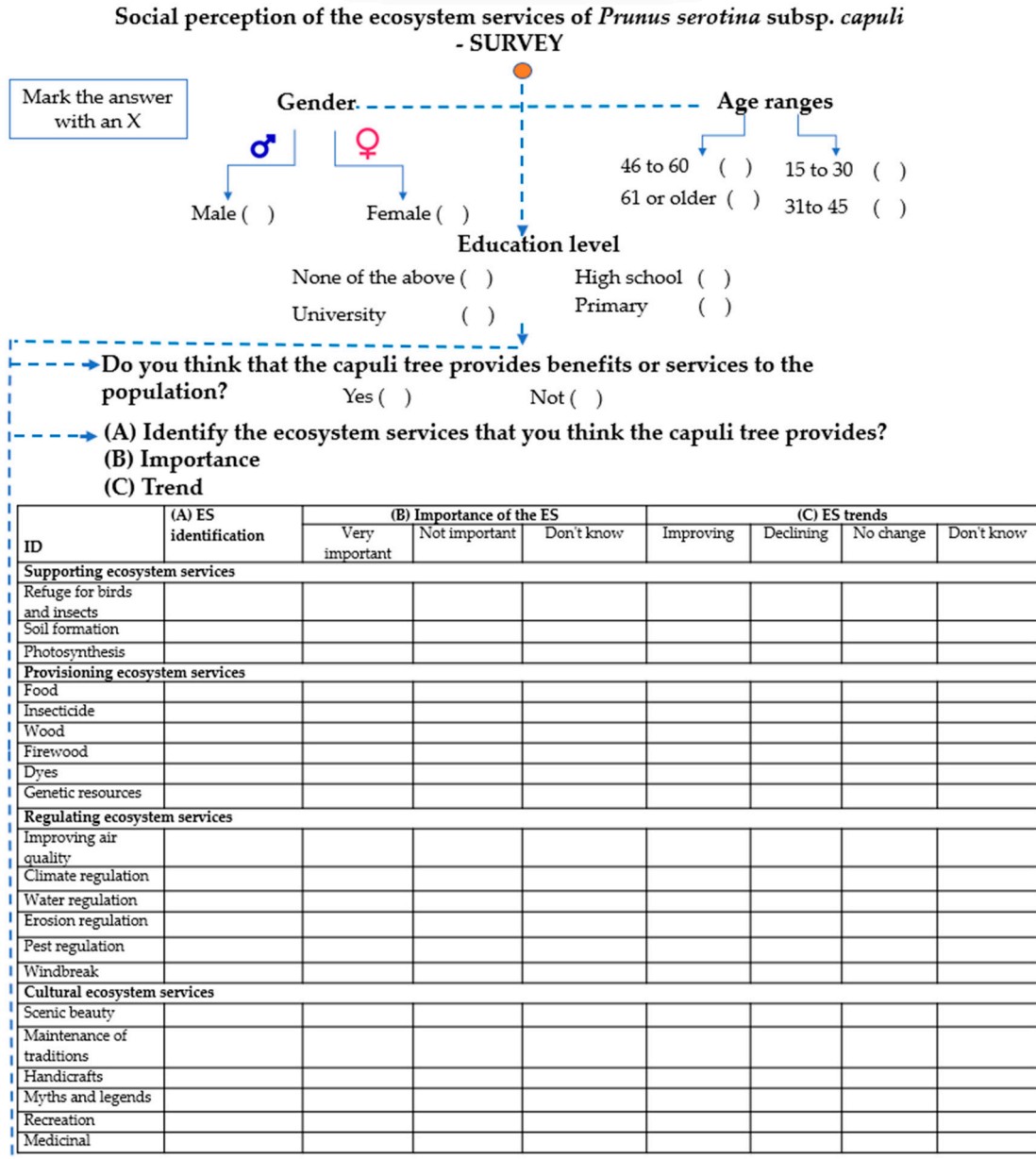

**Figure A1.** Interview format to determine the social perception of the ES provided by the species *Prunus serotina*.

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
