# Peer review of "Social Perception of the Ecosystem Services of Prunus serotina subsp. capuli in the Andes of Ecuador"

_land, doi:10.3390/land12051086_

Round 1
Reviewer 1 Report (Previous Reviewer 1)
I do not have other comments.
I do not have other comments.
Author Response
Greetings dear reviewer, we hope you are very well. Thank you for your comments.
Reviewer 2 Report (Previous Reviewer 3)
Social perception of the ecosystem services of Prunus serotina subsp. capuli in the Andes of Ecuador
Abstract: Line 24 change for “R-studio”.
Introduction: Line 53 change for “2.0”.
Line 69 delete “provinces”.
Line 78, use comma not “;”. Please revise the rest of the manuscript for this mistake (e.g. line 103).
Use the acronym for ES (ecosystem service) in all the text. Name in the first appearance, and then, the acronym.
Line 109 change work for research. Here you need to add some objective for the full research (use the title as key words) before the questions. Also refer that this research will be based on surveys and data bases obtained in the field. Right?
Line 139, you need to use comma for the thousands? Check the norms for authors please, and change in all the text.
Figure 2 must be improved in quality. First check if you use the same caption, and then, remove the outside line. Third, add some (A), (B), and (C), and explain in the caption. Remove the provinces, and explain the colours in the caption. For ages, use >61, 31-45, etc. In the caption explain the units (years-old). The same for the other figures.
I wonder is some statistical analyses can be conducted to compare the provinces, or the different treatments. Otherwise is descriptive and you can not support the conclusions.
3.4 is Supporting services.
Figure 4 presented very low captions. Must be improved. The ES must be described in the caption, and in the graphs, you can use acronyms (e.g. WR = water regulation). Idem for Figure 5, we need to simplify the figure, To much text.
Conclusions must include few sentences with the NEW knowledge for the Science. This is not a summary of your results. Otherwise, change the title for “Final Remarks”.
Author Response
Greetings dear reviewer, we hope you are very well. Thank you for your comments.

This manuscript is a resubmission of an earlier submission. The following is a list of the peer review reports and author responses from that submission.
Round 1
Reviewer 1 Report
This paper determined the knowledge, perception about the importance of the ecosystem services of Prunus serotina subsp. capuli, as well as the factors that influence their knowledge and perceptions in the rural areas of three study areas (Chimborazo, Tungurahua and Cotopaxi) of the central Andes of Ecuador. The general framework and statistical analysis are correct. But the paper still has room for improvement.
1. Introduction. Has there been any similar research before? How does this paper reflect its importance and necessity compared with previous research? You can add a description to the introduction to explain these issues. Additionally, the research on the social perception of ecosystem services should be appropriately supplemented.
2. The main content of this paper is to discuss the ecosystem services provided by Prunus serotina species. Then, whether there is encroachment of Prunus serotina species on other ecological lands (such as forests and grasslands) in the study area, whether this will have an impact on local ecosystem services, and how to measure the impacts? Further consideration is recommended
3. As pointed out in comment 2, if the importance and trend of the ecosystem services provided by Prunus serotina species is based on interview data, will it be significantly different from the actual situation? How can we judge its guidance for the real society? This should be further explored in the discussion.
4. It seems that the importance and trend of the specific ecosystem services provided by Prunus serotina species are not explained in detail in the paper. It is suggested to add a supplementary explanation.
5. As mentioned in the comment 4, the selection of interviewees will inevitably have an impact on the research results. What principles are used to select interviewees in the research?
6. It was mentioned that due to differences in knowledge level and cognition, interviewees may have large individual differences in their evaluation on the type of ecosystem services provided by Prunus serotina species. How to deal with this problem in the research and verify the rationality and scientific nature of the experiment? It is suggested to be further discussed.
Reviewer 2 Report
The article presented by the authors is well constructed. It is innovative and suggestive. I have no comments to make on this clear and well-written paper. I think it is fantastic. Congratulations to the authors.
Reviewer 3 Report
Abstract: Line 23: remove (excel 2019), left: “...interview data was analysed to obtain thee relationships among variables by the Spearman correlation...”
Line 24: remove version 4.1.1. (let this information in methods).
Line 25: change showed by indicated.
Introduction: Line 80: The lack of information is not enough to justify one study. Try to improve the introduction with the advantages for the people and policy-makers of this new knowledge, how thy can contribute to the management, conservation and planning in the region. Add other examples around the World, where the knowledge allowed to manage and conserve the species.
Methods: The spreadsheet of the Fig 2 must be moved to the Annex. The rationale of the study can remain in the text.
Results: Style of Fig 3 and 4 must be the same. Remove the vertical and horizontal lines in the Graph. Use acronyms (simplify the graph) for ES types. You can remove the numbers in both graphs. I wonder if you can introduce some statistics in these graphs? There are differences, but these differences are significant among communities?
Same comments for Fig 5. Remove the frameworks and lines (vertical and horizontal).
Conclusions are not conclusions, are a summary of the results. The conclusions must reflect the new knowledge obtained for the science. Please consider to improve this section.